# Conservation Lessons from the Study of North American Boreal Birds at Their Southern Periphery

**Joel Ralston [1],\* and William V. DeLuca [2]**

[1]  Department of Biology, Saint Mary's College, Notre Dame, IN 46556, USA
[2]  National Audubon Society, Amherst, MA 01075, USA; william.deluca@audubon.org
\*  Correspondence: jralston@saintmarys.edu

**Abstract:** Many North American boreal forest birds reach the southern periphery of their distribution in the montane spruce–fir forests of northeastern United States and the barren coastal forests of Maritime Canada. Because the southern periphery may be the first to be impacted by warming climates, these populations provide a unique opportunity to examine several factors that will influence the conservation of this threatened group under climate change. We discuss recent research on boreal birds in Northeastern US and in Maritime Canada related to genetic diversity, population trends in abundance, distributional shifts in response to climate change, community composition, and threats from shifting nest predators. We discuss how results from these studies may inform the conservation of boreal birds in a warming world as well as open questions that need addressing.

**Keywords:** range periphery; spruce–fir forests; climate change; range shift; community dynamics

---

## 1. Introduction

The North American boreal forest covers approximately 600 million ha of land across northern United States and Canada and is home to up to 3 billion breeding birds of more than 300 species [1]. Still a relatively undisturbed ecosystem across much of its extent, the boreal forest is perhaps the most productive ecosystem for birds in North America [2]. Yet the boreal forest faces considerable threats from various environmental pressures—perhaps the most important of which is climate change. High-latitude ecosystems such as boreal forests are expected to experience the greatest temperature changes in the coming decades [3], increased frequency of wildfires [4], and major changes in biodiversity [5]. As a result, boreal birds are among the most threatened bird communities by ongoing climate change [6,7]. This highlights the importance of understanding the ecology of boreal populations currently being impacted by climate change, and using that information to inform conservation action across the boreal.

The boreal forest reaches its southern and eastern peripheries in the low-lying wetlands and montane forests of northeastern United States and in coastal barren forests of maritime Canada [8,9] (Figure 1). These peripheral populations are likely to be the first to experience the ecological consequences of ongoing anthropogenic climate change. Southern periphery high-elevation populations are expected to be the first to be extirpated as habitats become unsuitable and species' distributions shift in response [10–12]. Additionally, as temperate species from further south shift into higher latitudes, southern boreal species will experience novel ecological interactions via competition, predation, and disease—each of which may impact population trends and community structure [12–15]. Lastly, southern boreal forests are impacted by recreation, land use change, development and disturbance associated with human populations at a much higher rate than the relatively undisturbed boreal forests further north [12,16–20]. For these reasons, we describe the southern boreal forest in northeastern United States and southern Canada as the current "battle ground" of environmental change for the

boreal forest. Studies of ecological change and the impacts of climate change in these peripheral populations may therefore provide insight into how climate change will continue to impact boreal forests more broadly in the years to come. Here, we review approximately a decade of research in the southern periphery of the boreal forest, and synthesize several conservation lessons as well as ongoing questions related to the impacts of climate change on this community.

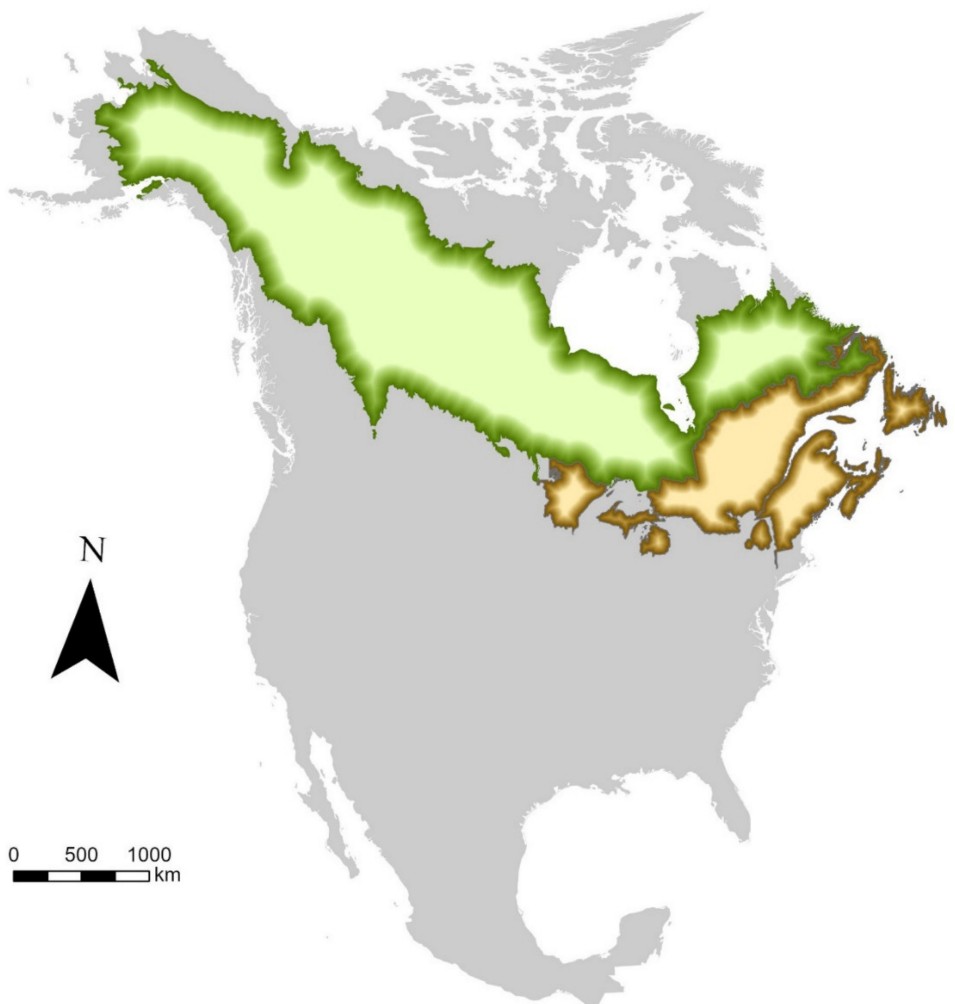

**Figure 1.** North American boreal forest (green) and a general approximation of its southern periphery (brown), where most of the studies reviewed here took place. The boreal forest and the southern periphery were identified by Bird Conservation Regions 4, 6, 7, 8, 12, and 14.

## 2. Peripheral Populations of Boreal Birds Are Genetically Unique, Threatened by Climate Change, and Declining

Because southern peripheral populations are geographically disjunct, they may represent an 'archipelago' of independently evolving populations isolated from gene flow and holding unique genetic diversity important to conserve under climate change [21,22]. In an analysis of microsatellite diversity, Ralston and Kirchman [11] found that southern peripheral populations of Blackpoll Warblers (*Setophaga striata*) in montane forests of northeastern United States may hold as much as 10% of the allelic richness in this species. An analysis of mitochondrial haplotypes across four species of boreal birds also found a periphery effect, with greater differences in genetic diversity between peripheral populations than between populations across the contiguous boreal forest [23]. This effect varied across species, with Blackpoll Warbler and Canada Jay (*Perisoreus canadensis*) showing a larger effect than Boreal Chickadee (*Poecile hudsonica*) and Yellow-bellied Flycatcher (*Empidonax flaviventris*), suggesting

the isolating effect of peripheral populations may vary across species [23]. As populations decline at the southern periphery of the boreal, it is likely that this unique genetic diversity will be eroded [11,24].

In addition to southern peripheral populations, Newfoundland, an eastern peripheral isolate, has been shown to contain genetically unique populations of several boreal bird species including Canada Jay [25,26], Boreal Chickadee [27], Blackpoll Warbler [28], Gray-cheeked Thrush (*Catharus minimus*) [29], and Fox Sparrow (*Passerella iliaca*) [30]. Genetic divergence on Newfoundland may be due to isolation during the Pleistocene in a refugium located at the now submerged Grand Banks area east of modern day Newfoundland [25,27], or due to limited modern gene flow between Newfoundland and continental populations [23,28].

In addition to unique intraspecific genetic diversity observed in several species, the southern boreal forest is also home to an endemic bird species. The Bicknell's Thrush (*Catharus bicknelli*) breeds in upland spruce–fir forests in Northeastern United States, Quebec, New Brunswick, and Nova Scotia [31], a distribution closely matching the 'southern periphery' of other boreal forest species described above. While intraspecific divergence in other eastern boreal species has been dated to the Holocene, divergence between Bicknell's Thrush and its northern sister species, Gray-cheeked Thrush, has been dated to the Pleistocene [23,32]. Pleistocene divergence in these species likely occurred in separate glacial refugia, and may be reinforced by habitat niche differentiation or competition [29,33]. While deep genetic divergence in this *Catharus* species complex is unique among eastern boreal species studied thus far [23], Fitzerald et al. [29] suggest the possibility that similar cryptic species breaks may exist for other boreal bird species at the southern and eastern peripheries.

As the climate continues to warm, it is these unique peripheral populations that will be first impacted. Ralston and Kirchman [11] modeled predicted changes in climate suitability for 15 boreal forest birds under several climate change scenarios across the southern boreal. Suitability for all species was predicted to shift northward (mean 934.0 km under the most severe scenario), with 12 of 15 species predicted to be extirpated (99–100% decrease in suitable area) in New York, Vermont, and New Hampshire by 2080, resulting in a significant decline in boreal bird diversity in this region. While these simple correlative models used only climatic variables, further analysis with more sophisticated models has similarly projected declines in the suitability of boreal birds in northeastern United States in the coming decades. McGarigal et al. [34] used climate, habitat, and land use variables to calculate and project changes in Landscape Capability, an index reflecting the ability of the landscape to meet species' breeding season natural history requirements [35]. There was a projected 64% decline in Landscape Capability for Blackpoll Warbler across northeastern United States by 2080, along with a predicted upslope shift in distribution greater than 100 m.

Impacts from climate change on bird distributions and abundances are already being observed in this region. In a resurvey of birds along an altitudinal gradient in New York state [36], Kirchman and Van Keuren [37] found the abundance weighted mean altitude shifted upslope an average 82.8 m ($n$ = 42 species) between 1974 and 2015. This result mirrors similar findings for long term upslope shifts in montane forest ecotones [38]. As climate change pushes species up in elevation, we expect populations to decline due to less area available at higher elevations [39], especially in the southern parts of their range where birds are already found near the tops of mountains. However, there appears to be variation in elevational shifts across sites and species. DeLuca and King [12] found that 9 out of 11 montane boreal species in the White Mountains shifted their lower elevational range boundary downslope on average 19 m between 1993 and 2009. This shift may have been a response to regrowth of spruce and fir following historic logging [12,40].

As climate change continues, boreal birds are declining. Continental-scale point count surveys typically used to assess long term population trends in birds are usually unsuitable for boreal species given the inaccessibility of their habitats [41–43], but the estimates that are available often show steep declines. Blackpoll Warblers, an emblematic bird of the boreal forest and historically perhaps the most abundant, has seen a 92% decline since the 1970s [44]. To estimate trends at the local and regional scales, numerous surveys exist in montane spruce–fir forests, or lowland boreal spruce bogs across

the southern boreal, operated by conservation organizations, national forests, parks, and wildlife refuges [45–51]. Several of these programs have now been operated for over two decades, and most are showing a general pattern of declining bird abundances across the southern boreal. Vermont Center for Ecostudies' Mountain Birdwatch has documented a significant 10 year decline in 7 of the 10 species that it monitors in montane forests in northeastern United States and Canada [51]. The only species monitored that significantly increased in the Mountain Birdwatch dataset was Black-capped Chickadee (*Poecile atricapilla*), indicating the movement of low-elevation species into high-elevation boreal habitats and potentially a change in community structure [51]. Similarly, Glennon et al. [50] found that boreal birds generally declined in occupancy at lowland boreal wetland sites in New York over a decade of observation, while non-boreal species were more likely to increase in occupancy. Further, non-boreal species were more likely to colonize sites that boreal species disappeared from, again indicating changes in community composition [50]. Ralston et al. [42] combined 16 of these local and regional survey datasets to estimate population trends for 14 boreal bird species across the northeastern and upper Midwestern United States. They found that four species commonly considered indicator species of boreal spruce–fir forests, Yellow-bellied Flycatcher, Bicknell's Thrush, Blackpoll Warbler, and Magnolia Warbler (*Setophaga magnolia*), each showed significant declines across the study region. Further, species identified as spruce–fir specialists were more likely to be declining than those which also used other habitat types, indicating that pressures driving population declines in abundance in birds may be particularly strong in boreal spruce–fir forests.

In addition to showing general declines in spruce–fir birds, Ralston et al. [42] demonstrated considerable regional variation in trends within species. In five of the nine species with sufficient data for analysis in northeastern and upper Midwestern sites, trends were significantly different between regions. In three species the direction of change differed. Canada Jay and Swainson's Thrush (*Catharus ustulatus*) each declined in the upper Midwest while increasing in abundance in the Northeast; Red-breasted Nuthatch showed the opposite pattern, increasing in the upper Midwest while declining in the Northeast [42]. Similarly, all species showed variation in trend estimates across surveys within each region. At a local scale, Glennon et al. [50] also showed variation among sites in boreal bird community dynamics. Boreal birds were more likely to persist at sites with a greater amount of open Northern Peatland compared to sites with Boreal Upland Forest (habitat classifications following [9]). However, it is unlikely that these results apply to boreal species more typical of Boreal Upland Forests such as Blackpoll Warbler and Bicknell's Thrush [50]. Identifying the habitat factors associated with stable or increasing trends in boreal birds at the southern periphery could be very important for conservation practitioners. Cooperation among stakeholders to extend previous analyses to a greater number of sites, boreal habitat types, and regions throughout the southern boreal could be tremendously helpful for the conservation of boreal birds more broadly. The importance of support for targeted boreal monitoring programs cannot be overstated. Current population estimates for boreal species based on road-based surveys have proven to be inaccurate and often underestimate boreal population estimates [43].

## 3. Species Differ in Their Responses to Changing Environments, Creating a Challenge for Community Level Conservation

Drivers of species distributions often vary across species. For boreal birds, the extent to which habitat and/or climate limit distributions has major implications for the conservation and management of those species. For example, if a species' distribution is primarily limited by climate, focusing on climate refugia preservation might be the preferred approach; if a species was primarily limited by habitat, management of habitat characteristics might be the optimal approach. However, a challenge of species-focused approach is that management plans designed to conserve a single species may have limited benefits for other threatened species in the community. Several recent studies in habitat associations of boreal birds have shown that species differ in their habitat requirements and in the environmental factors that drive their occupancy [12,50,52–54]. Climatic factors were

shown to be generally more important than landscape factors (i.e., connectivity, human footprint) to occupancy dynamics in low-elevation boreal birds, though species differed in what climate factors were most important [54]. Additionally, indirect effects of climate as mediated by species interactions (i.e., insect prey abundance) and vegetation structure may be important to dynamics of these species on relatively short and long time scales, respectively [54]. Similarly, a large proportion of the impact of climate on the distribution of montane boreal birds along an elevation gradient was indirect, mediated by vegetation, and again there was great variation in the strength and direction of direct in indirect effects of climate across species [52]. Together, these results demonstrate variation in habitat associations across boreal birds. These differences lead to regional variation in the structure of boreal bird communities along the southern periphery of the boreal, and ongoing climate change may drive changes in community composition [50,53].

While it has been previously demonstrated that species associations may be weak in southern boreal forests, creating challenges to community level conservation [55], recent studies on boreal birds suggest that ongoing climate change may exacerbate these challenges by driving changes in community composition [50,52–54]. For example, designing forest management practices to benefit a species of conservation concern, Bicknell's Thrush [56,57], may have relatively smaller impact on other declining species such as Blackpoll Warbler or Canada Jay [42] that appear to be more strongly and directly influenced by climate variables [52,53]. While protecting climate refugia where boreal communities can persist may be an important conservation strategy [6,50], conserving the boreal community in its current form will become increasingly challenging. Species of conservation concern may be shuffled into non-analogous communities not currently present in the landscape, requiring unique conservation strategies [58]. Conservation practitioners focused on boreal species will therefore need to clearly define whether the objectives of management efforts are to protect individual species or larger assemblages of boreal birds.

## 4. As Communities Reshuffle, Species Face New Biotic Interactions—Some of Which May Have Unexpected Outcomes

As climate change drives changes in community composition, species will be faced with novel ecological interactions and challenges. For boreal birds at their southern periphery, increased contact with congeneric competitors may be increasingly important with climate change. Southern limits of northern species and, by extension, low-elevational limits of high-elevation species, are often determined by biotic interactions, mainly competition [59–61]. Many of the boreal species discussed thus far have more southerly or low-elevation relatives with which they compete [61,62], and that may be shifting into historically boreal sites [12,50,51]. In recent decades, as Bicknell's Thrush has been declining, the congeneric Swainson's Thrush has been shifting upslope and increasing in abundance, followed by Hermit Thrush (*Catharus guttata*), a historically low-elevation competitor species which can now be observed at many high-elevation sites [12,37,51,62]. Similarly, Boreal Chickadee may be being outcompeted and replaced by Black-capped Chickadees in many places [51]. In the White Mountains, warbler species typically associated with lower-elevation mixed forests shifted on average upslope into boreal habitats while high-elevation boreal warblers such as Blackpoll Warbler, Yellow-rumped Warbler (*Setophaga coronata*), and Nashville Warbler (*Oreothlypis ruficapilla*) have shifted downslope [12], increasing the likelihood of competition among these species. While warblers are a classic example of foraging niche differentiation to reduce competition [63], it is unclear how these high-elevation boreal species will respond to novel competitors.

Another possibility, as climate change pushes closely related species into contact, is hybridization. Secondary contact of related species following environmental change can result in widespread introgression, loss of genetic diversity from a parent species, and even speciation reversal or the collapse of two lineages into one [64–66]. This can create additional conservation concern, especially for rare species that come into contact and hybridize with a more abundant species [67]. However, hybridization can be a tricky conservation topic, as it may not always be clear whether hybrids are

naturally occurring or the result of anthropogenic disturbance [68]. As climate change reshuffles species distributions and community composition, it is possible that there might be increased opportunities for hybridization, especially at range peripheries, where species' distributions abut one another. FitzGerald et al. [32] recently documented hybridization between Bicknell's Thrush and its northern sister species Gray-cheeked Thrush. The hybrid was phenotypically Gray-cheeked Thrush and was captured in southern Labrador within the Gray-cheeked Thrush's distribution, but held a Bicknell's Thrush mitochondrial haplotype [32]. Further analysis of both species has revealed low levels of genetic admixture within Bicknell's Thrush populations [29]. While hybridization between these species has been suggested in the past [69], this is the first confirmed case. While this likely represents a natural case of hybridization, it is unclear whether anthropogenic climate change may increase the frequency of hybridization over time, either through changes in distribution or migratory behavior (i.e., late-migrating Gray-cheeked Thrush, breeding with Bicknell's Thrush, as they migrate through Bicknell's Thrush territories further south) [32].

An important biotic interaction currently impacting peripheral boreal populations is the climate-driven introduction and spread of Red Squirrels (*Tamiasciurus hudsonicus*) an avian nest predator. Red Squirrels may be shifting downslope, opposite expectations given climate change [70] and are impacting nesting success of boreal birds in montane habitats [71]. Nest survival in Blackpoll Warblers and Bicknell's Thrush is highest at elevations with the lowest Red Squirrel abundance, and in years with lower squirrel abundance [71]. Similarly, the introduction and spread of Red Squirrels on Newfoundland has been coincident with the steep decline in Newfoundland Gray-cheeked Thrush populations [15,72], and Gray-cheeked Thrush were more likely to be detected on point counts where Red Squirrels were absent [32]. An interesting aspect of this is that Red Squirrels appear to be co-distributed with boreal birds elsewhere across the contiguous boreal forest with seemingly lower impacts on population abundance than at the southern and eastern peripheries of the boreal. This suggests that biotic interactions introduced through climate change can have unexpected consequences. Further work needs to be performed to understand how differing ecological contexts in peripheral populations and throughout the boreal influence the outcomes of these interactions.

## 5. Conclusions

Boreal forests are one of the most threatened ecosystems under climate change. At the southern periphery of the boreal forest, where climate change is already having an impact, boreal birds are declining in abundance and genetic diversity [24,42,51], and boreal community composition is changing [50,53]. We can take several conservation lessons from the studies of these peripheral boreal bird communities. First, species-specific responses to climate change may require individualized management plans, or flexible conservation that accounts for and is flexible to non-analogous communities. Second, as communities reshuffle, new biotic interactions are likely, and their outcomes may be unique to this new ecological context, providing additional challenges for conservation. Continued and expanded monitoring of boreal forest bird communities will be essential to detect further declines and understanding consequences of novel species assemblages. Future coordination of research and monitoring efforts across boreal regions and habitat types, and among conservation stakeholders, will help us to understand why trends in boreal birds vary regionally and what factors promote local increases in abundance and will be crucial for the conservation of boreal birds.

**Author Contributions:** Conceptualization, J.R. and W.V.D.; writing—original draft preparation, J.R.; writing—review and editing, W.V.D. All authors have read and agreed to the published version of the manuscript.

**Funding:** This research received no external funding.

**Acknowledgments:** This manuscript was developed from a talk given by the authors in the "Conservation and Management of Boreal Birds in a Changing Climate" symposium organized by Drs. D. Stralberg, S. Matsuoka, and J. Tremblay at the 2019 meeting of American Ornithological Society. We thank the organizers and the participants in that symposium for feedback and discussion on the topic. Additionally, we thank our many collaborators over the previous decade who have taught us a great deal about boreal birds.

**Conflicts of Interest:** The authors declare no conflict of interest.

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
