# Peer review of "Conservation Lessons from the Study of North American Boreal Birds at Their Southern Periphery"

_diversity, doi:10.3390/d12060257_

Round 1
Reviewer 1 Report
I think this is a well researched and important topic that will hopefully bring more attention to an overlooked area in need of conservation. However, given that this is a perspective, I feel more attention needs to be given to the delivery of the ideas and have the authors provide a better synthesis of the research to highlight the most critical aspects. Also, the authors should not be afraid to use a bit more forceful/intentional language. For example, the last sentence of the manuscript reads: “Future research on why population trends in boreal birds vary regionally and what factors promote local increases in abundance could be beneficial for the conservation of boreal birds.” I think ‘could be beneficial’ is grossly understating the importance. Lastly, it feels like some of the paragraphs are simply lists of examples with a concluding sentence stating some form of ‘more work needs to be done’. I feel if these suggestions are addressed, the manuscript would be much more impactful and garner more attention.
Some minor comments:
If allowed, I think a map highlighting the periphery of the boreal (southern and eastern isolate) that is referred to in this manuscript would be useful to many readers.
Ln 16: replace first ‘and’ with a comma
Ln 27: citation needed for ‘greatest temperature changes in the coming decades’
Ln 41 delete first ‘and’
Ln 158 awkward sentence, maybe missing a word
Ln 163 insert comma or delete ‘importantly’
Ln 160-174 This section could be re-worked to read less like a litany of examples. The last sentence also feels out of place.
Ln 231 detected
Ln 247 detect
Author Response
Reviewer 1
I think this is a well researched and important topic that will hopefully bring more attention to an overlooked area in need of conservation. However, given that this is a perspective, I feel more attention needs to be given to the delivery of the ideas and have the authors provide a better synthesis of the research to highlight the most critical aspects. Also, the authors should not be afraid to use a bit more forceful/intentional language. For example, the last sentence of the manuscript reads: “Future research on why population trends in boreal birds vary regionally and what factors promote local increases in abundance could be beneficial for the conservation of boreal birds.” I think ‘could be beneficial’ is grossly understating the importance. Lastly, it feels like some of the paragraphs are simply lists of examples with a concluding sentence stating some form of ‘more work needs to be done’. I feel if these suggestions are addressed, the manuscript would be much more impactful and garner more attention.
Thank you for your suggestions. While we keep our call for future studies in many cases, we also use stronger language as suggested, and try to be more direct about or conclusions, especially in the Conclusions section.
Some minor comments:
If allowed, I think a map highlighting the periphery of the boreal (southern and eastern isolate) that is referred to in this manuscript would be useful to many readers.
Map figure added.
Ln 16: replace first ‘and’ with a comma
“habitat selection and…” removed
Ln 27: citation needed for ‘greatest temperature changes in the coming decades’
Citation added.
Ln 41 delete first ‘and’
Deleted as suggested
Ln 158 awkward sentence, maybe missing a word
Editied to: Several recent studies in habitat associations of boreal birds have shown that species differ in their habitat requirements and in the environmental factors that drive their occupancy.
Ln 163 insert comma or delete ‘importantly’
‘importantly’ removed as suggested
Ln 160-174 This section could be re-worked to read less like a litany of examples. The last sentence also feels out of place.
Section reworked, and last sentence deleted.
Ln 231 detected
Corrected
Ln 247 detect
Corrected
Reviewer 2 Report
This is a review/overview paper and therefore my comments are very minor. I appreciate this paper for its simplicity and clarity and appreciate the role it plays in the body of literature concerning boreal birds.
Specific comments (by line number):
23 - change over to more than
25 - remove s from ecosystems
39 - phrasing makes it sound as if competition, predation, and disease are novel which I assume is not what you mean, perhaps rephrase to say that species will experience novel interactions VIA competition, predation, etc.
58 - can you provide a brief explanation of what a periphery effect is for a novice who does not know what Fst values are?
106 - change to continental scale
108 - although I do not object to this phrasing, I have received negative feedback from reviewers about using words like grim to describe estimates and trends because they sound too value-laden. I leave it up to you but raise it with respect to how some readers may respond.
125 - should read across THE northeastern
155 - should have an apostrophe on species in this context; a species' distribution
158 - do you mean a challenge OF a species-focused approach is...?
176 - remove "the" before community
184 - into instead of in to
231 - change detect to detected
241 - remove and
247 - change detected to detect or detecting
385 - remove change from title of paper
Author Response
This is a review/overview paper and therefore my comments are very minor. I appreciate this paper for its simplicity and clarity and appreciate the role it plays in the body of literature concerning boreal birds.
Thank you for your comments and edits. All suggested changed were made.
Specific comments (by line number):
23 - change over to more than
Changed as suggested.
25 - remove s from ecosystems
Corrected.
39 - phrasing makes it sound as if competition, predation, and disease are novel which I assume is not what you mean, perhaps rephrase to say that species will experience novel interactions VIA competition, predation, etc.
Changed as suggested.
58 - can you provide a brief explanation of what a periphery effect is for a novice who does not know what Fst values are?
Replaced “greater Fsts values” with more general “greater differences in genetic diversity”
106 - change to continental scale
Changed as suggested.
108 - although I do not object to this phrasing, I have received negative feedback from reviewers about using words like grim to describe estimates and trends because they sound too value-laden. I leave it up to you but raise it with respect to how some readers may respond.
Changed to ”…estimates that are available often show steep declines.”
125 - should read across THE northeastern
Changed as suggested.
155 - should have an apostrophe on species in this context; a species' distribution
Corrected.
158 - do you mean a challenge OF a species-focused approach is...?
Corrected.
176 - remove "the" before community
Corrected.
184 - into instead of in to
Corrected.
231 - change detect to detected
Corrected.
241 - remove and
Changed as suggested.
247 - change detected to detect or detecting
Corrected.
385 - remove change from title of paper
Corrected.